# Early Inhibition of Phosphodiesterase 4B (PDE4B) Instills Cognitive Resilience in APPswe/PS1dE9 Mice

**DOI:** 10.3390/cells13121000

**Published:** 2024-06-08

**Authors:** Ben Rombaut, Melissa Schepers, Assia Tiane, Femke Mussen, Lisa Koole, Sofie Kessels, Chloë Trippaers, Ruben Jacobs, Kristiaan Wouters, Emily Willems, Lieve van Veggel, Philippos Koulousakis, Dorien Deluyker, Virginie Bito, Jos Prickaerts, Inez Wens, Bert Brône, Daniel L. A. van den Hove, Tim Vanmierlo

**Affiliations:** 1Department of Neuroscience, Biomedical Research Institute, Faculty of Medicine and Life Sciences, Hasselt University, 3500 Hasselt, Belgium; ben.rombaut@uhasselt.be (B.R.); m.schepers@maastrichtuniversity.nl (M.S.); a.tiane@maastrichtuniversity.nl (A.T.); femke.mussen@uhasselt.be (F.M.); lisa.koole@maastrichtuniversity.nl (L.K.); sofie.kessels@uhasselt.be (S.K.); chloe.trippaers@uhasselt.be (C.T.); ruben.jacobs@maastrichtuniversity.nl (R.J.); emily.willems@uhasselt.be (E.W.); lieve.vanveggel@uhasselt.be (L.v.V.); p.koulousakis@maastrichtuniversity.nl (P.K.); inez.wens@uhasselt.be (I.W.); bert.brone@uhasselt.be (B.B.); 2Department Psychiatry and Neuropsychology, Mental Health and Neuroscience Institute (MHeNs), Division Translational Neuroscience, Maastricht University, 6229 ER Maastricht, The Netherlands; jos.prickaerts@gmail.com (J.P.); d.vandenhove@maastrichtuniversity.nl (D.L.A.v.d.H.); 3University MS Center (UMSC) Hasselt, 3900 Pelt, Belgium; 4Department of Immunology and Infection, Biomedical Research Institute, Hasselt University, 3500 Hasselt, Belgium; 5Department of Psychiatry and Behavioral Sciences, Johns Hopkins University School of Medicine, Baltimore, MD 21218, USA; 6Department of Internal Medicine, Maastricht University Medical Center+ (MUMC+), 6229 ER Maastricht, The Netherlands; kristiaan.wouters@maastrichtuniversity.nl; 7Cardiovascular Research Institute Maastricht (CARIM), Maastricht University, 6229 ER Maastricht, The Netherlands; 8UHasselt, Cardio & Organ Systems (COST), BIOMED, Agoralaan, 3590 Diepenbeek, Belgium; dorien.deluyker@uhasselt.be (D.D.); virginie.bito@uhasselt.be (V.B.); 9Department of Psychiatry, Psychosomatics and Psychotherapy, University of Wuerzburg, 97070 Wuerzburg, Germany

**Keywords:** Alzheimer’s disease, cyclic nucleotide phosphodiesterases type 4B, synapse, microglia, cognition

## Abstract

Microglia activity can drive excessive synaptic loss during the prodromal phase of Alzheimer’s disease (AD) and is associated with lowered cyclic adenosine monophosphate (cAMP) due to cAMP phosphodiesterase 4B (PDE4B). This study aimed to investigate whether long-term inhibition of PDE4B by A33 (3 mg/kg/day) can prevent synapse loss and its associated cognitive decline in APPswe/PS1dE9 mice. This model is characterized by a chimeric mouse/human APP with the Swedish mutation and human PSEN1 lacking exon 9 (dE9), both under the control of the mouse prion protein promoter. The effects on cognitive function of prolonged A33 treatment from 20 days to 4 months of age, was assessed at 7–8 months. PDE4B inhibition significantly improved both the working and spatial memory of APPswe/PSdE9 mice after treatment ended. At the cellular level, in vitro inhibition of PDE4B induced microglial filopodia formation, suggesting that regulation of PDE4B activity can counteract microglia activation. Further research is needed to investigate if this could prevent microglia from adopting their ‘disease-associated microglia (DAM)’ phenotype in vivo. These findings support the possibility that PDE4B is a potential target in combating AD pathology and that early intervention using A33 may be a promising treatment strategy for AD.

## 1. Introduction

Alzheimer’s disease (AD) is a leading cause of dementia and, due to the increase in average life expectancy and the growing worldwide numbers of individuals aged 65 or older, we are projected to see a doubling of AD patients by 2050 [1,2]. Late-stage AD patients experience loss of basic bodily functions such as walking or swallowing, requiring around-the-clock care as the disease progresses to fatality. Early symptoms are associated with cognitive decline due to neuronal loss in brain regions responsible for memory, such as the hippocampus and the prefrontal cortex. However, disease progression varies on a case-by-case basis and can include changes in mood, personality, or behavior.

AD pathology is divided into three distinct phases: preclinical, prodromal with associated mild cognitive impairment (MCI), and a distinctive dementia phase. In individuals with MCI, altered brain imaging and cerebrospinal fluid biomarkers have been linked to progression to dementia [3]. Once the dementia stage is reached, existing treatments can delay but cannot halt the disease progression. A strategy is needed to target the preclinical phase and prevent neurological damage and mental deterioration [4,5,6]. 

Synapse loss is a key component of the pathophysiology of AD and occurs before the onset of clinical symptoms [7]. Synapses are essential for neuronal function and cognitive performance, and synapse loss strongly correlates to cognitive decline [8,9]. Suppressing synapse loss at the earliest possible point should be an effective treatment strategy [10]. Microglia, the resident immune cells of the central nervous system (CNS), are responsible for brain homeostasis and synaptic pruning [11]. The dysregulation of synaptic pruning is believed to be a key mechanism in AD, leading to premature excessive synapse loss. Recent work has highlighted that microglia play an active role in synapse loss and the subsequent induction of cognitive decline in early stages of AD [12,13,14].

In its dementia phase, AD is classically characterized by multiple major neuropathological hallmarks, among which are sustained neuroinflammation and amyloid-β (Aβ) accumulation and plaque formation [15]. Research focused on Aβ has laid the foundation for the amyloid cascade hypothesis and is associated with the emergence of amyloid plaques, elevated oxidative stress, synaptic dysfunction, and neuronal loss [16,17]. Central to the amyloid hypothesis is the amyloid precursor protein (APP) and its downstream processing. Mutations in the genes encoding APP or the γ-secretase complex proteins presenilin 1/2 (PSEN1/2) can cause a shift toward the amyloidogenic pathway, causing the generation and aggregation of the neurotoxic peptide Aβ in the CNS [18]. In addition to mediating synaptic plasticity, microglia also prevent the toxic aggregation of Aβ by consistently clearing the peptide [19]. 

3′-5′-cyclic adenosine monophosphate (cAMP) is an important second messenger for microglial functioning, playing a role in the regulation of inflammation and phagocytosis [20,21,22,23]. In particular, cAMP blocks phagocytosis and reduces synaptic pruning by microglia [21,24]. Boosting cAMP enhances neuroplasticity and improves cognition in neurodegenerative conditions [25]. Phosphodiesterases (PDEs) are regulators of cAMP levels, by catalyzing their hydrolysis. PDEs comprise eleven families (PDE1-11), of which PDE4, 7, and 8 are responsible for the catabolic pathway of regulating cAMP levels specifically [26]. The PDE4 family is the most prominently expressed cAMP-specific PDE family in immune cells. Out of the four different PDE4 subtypes (PDE4A, B, C, and D), PDE4B is the most prevalent in microglia, rendering PDE4B an interesting therapeutic target to reduce neuroinflammation and to temper phagocytosis [27]. The anti-inflammatory properties of PDE4B inhibition have led to the development of selective inhibitors, such as A33 [28]. 

PDE4B- and general PDE- inhibition has been the subject of multiple studies aiming to intervene in neuropathologies, such as traumatic brain injury, AD, and spinal cord injury [29,30,31,32,33,34]. However, none of these studies have investigated the effect of restoring microglia functioning at an early stage. This study investigates elevating cAMP by the use of A33, a PDE4B inhibitor, in microglia in order to counteract synapse loss and improve cognitive function. In a combined study of in vivo experimentation and in vitro assays, we show here that PDE4B inhibition alters microglial functioning in vitro and improves synapse density and cognitive function in APPswe/PS1dE9 mice.

## 2. Materials and Methods

### 2.1. Animals 

APPswe/PS1dE9 (Jackson Laboratory, Bar Harbor Maine, USA) express a chimeric mouse/human APP with the Swedish mutation and human PSEN1 lacking exon 9 (dE9), both under the control of the mouse prion protein promoter. Mice of this model and Pde4b^−/−^ and mice with heterozygous expression of enhanced green fluorescent protein (eGFP) under the control of microglia-specific C-X-C chemokine receptor 1 (Cx_3_xr_1_^eGFP/+^ mice) were bred in-house and genotyped by PCR analysis of toe biopsies. Pde4b^−/−^ mice were kindly provided by Prof. Dr. O.N. Viacheslav (University Medical Center Hamburg-Eppendorf, German Center for Cardio-vascular Research) and Prof. Dr. M. Conti (University of California) and Cx3cr1^eGFP/eGFP^ mice were obtained from the European Mouse Mutant Archive (EMMA) institute with the approval of Steffen Jung (Weizmann Institute of Science) [35]. Wild-type mice (WT) bred on a C57BL/6 background (Inotiv) were bred in-house. All animals were housed in a temperature- and humidity-controlled environment (21–22 °C) with a normal day/night rhythm (lights on at 7 a.m., lights off at 7 p.m.) and ad libitum access to food and water. Transgenic female APPswe/PS1dE9 mice used for the in vivo experiment were housed individually in standard cages on sawdust bedding at the age of 7 months. They were kept under an inverse day/night rhythm (lights off at 7 a.m., lights on at 7 p.m.) to allow behavioral testing in their active (dark) phase during the day. The experiments were conducted in accordance with the guidelines of EU Directive 2010/63/EU on the protection of animals used for scientific purposes. All experiments were approved by the local ethical committee for animal experiments at Hasselt University (protocol IDs 202358B, 202244, 202132B, 202105, 202076B). Mice were housed and tested in the same room. A radio, which was playing softly, provided background noise in the room.

### 2.2. Primary Microglia Isolation 

Primary microglia for the phagocytosis assay were isolated from WT pups (postnatal day [P] 0–3) using the shake-off method described previously [26]. Briefly, meninges-free cerebral cortices were mechanically dissociated and chemically digested using 3 U/mL papain (Merck, Darmstadt, Germany, P4762) to obtain a cell suspension. After 4 days in culture, culture medium (1% Penicillin/Streptomycin (P/S)-containing DMEM (Gibco, Brussels, Belgium) supplemented with 10% fetal calf serum (FCS, Gibco)) was enriched with 1/3 L929-conditioned medium to generate microglia-enriched glial cultures. Microglia were obtained by using orbital shaking and seeded onto the well plate at the desired density. Experiments started 24 h after seeding. 

Cx_3_cr_1_^eGFP/+^ mice [35] bred on a C57BL/6 background were used for the surveillance experiments. Microglia were isolated from the cortices of pups (P21) by magnetic activated cell sorting (MACS), as described previously [36]. Briefly, cortices were harvested following cervical dislocation. Tissues were collected in cold DMEM (Gibco) and mechanically dissociated. The cell suspension was incubated for 30 min with papain (16 U/mg)/DNaseI solution at 37 °C and subsequently filtered through a 70 μm strainer with 1% P/S-containing DMEM to obtain a single-cell suspension. The cells were resuspended in 30% stock isotonic Percoll (Cytiva, Hoegaarden, Belgium 17089101) and underlaid with 70% Percoll solution. The 70%/30% density gradient was centrifuged at 500× *g* with brake 0 and acceleration 4 for 25 min to obtain a mononuclear cell layer. The cells were resuspended in MACS buffer (1× phosphate-buffered saline [PBS], 2% FBS, 200 mM EDTA) and incubated with CD11b microbeads (Miltenyi Biotec, Gladbach, Germany) for 15 min at 4 °C. The microglia were isolated using MS Columns (Miltenyi Biotec) and resuspended in culture medium (DMEM with 1% P/S and 10% FCS).

### 2.3. Synaptosome Isolation and Labeling

Synaptosomes were isolated from the brains of adult WT C57BL/6 mice. Brains were mechanically homogenized with the TissueRuptor (Qiagen, Hilden, Germany) in Syn-PER Synaptic Protein Extraction Reagent (87793, ThermoFisher Scientific, Geel, Belgium) supplemented with a protease (05892970001, Roche, Machelen, Belgium) and a phosphatase (04906837001, Roche) inhibitor cocktail. Synaptosomes were isolated by gradient centrifugation with Syn-PER Synaptic Protein Extraction Reagent according to the manufacturer’s instructions. Next, the synaptosomes were labeled with pHrodo dye (P3660, Invitrogen ThermoFisher, Geel, Belgium) according to the manufacturer’s instructions. Synaptosomes were stored at −80 °C in aliquots and every experiment was performed using synaptosomes from one single batch isolation.

### 2.4. Phagocytosis Assay

Microglia were seeded on a Poly-L-Lysine (PLL; 50 μg/mL, 25988-63-0, Sigma-Aldrich) pre-coated 24-well plate (200,000 cells/well) and after 24 h, they were pre-treated with 0.1% dimethylsulfoxide (DMSO) or 1 μM A33 (915082-52-9, Sigma-Aldrich, Overijse, Belgium) (IC50 PDE4B: 27 nM [28]) for the duration of 1 h. Afterwards, the cells were stimulated with pHrodo-labeled synaptosomes for 40 min. Images were obtained every 2 h, for 24 h, starting directly after the administration of the pHrodo-labeled synaptosomes with the Incucyte S3 Live-Cell Analysis System (Sartorius, Göttingen, Germany). Images were processed using Fiji ImageJ, and the area covered by pHrodo-positive cells, total cell area, and mean gray intensity were calculated. Afterward, the percentage of phagocytosing cells was calculated by dividing the pHrodo-positive area by the total cell area. The intensity was measured and normalized to the total cell area.

### 2.5. qPCR

Primary microglia isolated from Pde4b^+/+^, Pde4b^+/−^, and Pde4b^−/−^ pups (P0–3) were seeded on a PLL pre-coated 24-well plate (250,000 cells/well) and allowed to attach for 24 h. They were lysed in QIAzol lysis reagent (79306, Qiagen) and RNA was isolated using an isopropanol precipitation protocol [37]. Subsequent cDNA synthesis was performed using qScript cDNA Synthesis Kit (95047, QuantoBio, Beverly, MA, USA) and qPCR was performed with a SYBR Green (Applied Biosystems, Waltham, MA, USA) reporter on the QuantStudio™ 3 Real-Time PCR System according to the respective manufacturers’ instructions. Primer pairs used for each gene are shown in Table 1.

### 2.6. Surveillance

Primary murine *Cx3cr1^eGFP/+^* microglia were seeded on collagen (2 μg/mL; 9007-34-5, Sigma-Aldrich) and Poly-D-Lysine (PDL; A38904-01, Gibco) pre-coated glass inserts (80826, Ibidi, Gräfelfing, Germany) at a density of 30,000 cells/well in culture medium. After seven days, cells were switched to TIC medium (DMEM/F-12 (Gibco) supplemented with penicillin (100 units/mL), streptomycin (100 μg/mL), insulin (5 µg/mL), sodium selenite (100 ng/mL), apo-transferrin (100 µg/mL), heparan sulphate (1 µg/mL), acetylcysteine (5 µg/mL), human TGF-B2 (2 ng/mL), ovine wool cholesterol (1.5 µg/mL), murine IL-34 (100 ng/mL), and L-glutamine (2 mM)) to induce ramification over 7 days.

Before imaging, the TIC medium was replaced by Krebs solution (pH 7.4, 150 mM NaCl, 6 mM KCl, 1.5 mM CaCl2, 1 mM MgCl2, 10 mM HEPES, and 10 mM glucose monohydrate) before imaging with the Zeiss LSM880 confocal microscope. Filopodia motility was assessed during a time series just before and following the administration of 100 µM IBMX (Cayman Chemicals, Ann Arbor, MI, USA) as a positive control [20], 100 nM A33, or vehicle (0.1% DMSO). The images were acquired using a 63× oil immersion objective and the Airyscan detector. The surveilled area was determined by measuring the area covered by filopodia in a defined region of interest with Fiji ImageJ 1.53.

### 2.7. In Vivo Experimental Design

Experimental design is shown in Figure 1. Mice were bred in the institute and genotyped before P10 to allow randomized subdivision into three groups of females (*n* = 15) according to genotype and treatment: (1) WT mice receiving vehicle (DMSO), (2) APPswe/PS1dE9 mice receiving vehicle (DMSO), and (3) APPswe/PS1dE9 mice receiving A33 treatment. Between the ages of 20 days to 4 months (window of synaptic elimination), animals received daily subcutaneous injections of either DMSO (VWR) (1:1000; vehicle) or A33 (3 mg/kg)) dissolved in DMSO, and, subsequently, in 0.5% methylcellulose and 2% Tween80 [38]. Immediately after the treatment window, at 120 days of age, 4 mice per group were sacrificed to isolate the brain for downstream *post-mortem* analysis (see ‘Immunohistochemistry’). This timepoint of *post-mortem* analysis is further referred to as t1. The remaining animals (*n* ≥ 10/group) were subjected to behavioral testing at 7 months of age, followed by their sacrifice and subsequent *post-mortem* analysis (t2).

### 2.8. Behavioral Testing

The object location task (OLT) was performed when the mice were 7 months of age, when cognitive decline starts [39]. The OLT was performed as described previously [40]. Briefly, animals were placed in a circular arena with a diameter and height of 40 cm. The back-half of the arena wall was made of polyvinyl chloride covered with white paper. Testing was performed during two trials of 4 min (trial 1 with symmetrically placed objects; trial 2 with one stationary and one moved object) with an inter-trial interval time of 5 h to assess long-term memory. Between sessions, the arena and objects were cleaned thoroughly with a 70% ethanol solution. The exploration time for the two objects was measured manually using self-designed object location task software. The measures were used to calculate the discrimination index (d2 = (time spent on moved object − time spent on stationary object)/time spent exploring in trial 2). A d2 significantly higher than 0 indicates a preference for the moved object, while a d2 that is not different from 0 indicates no preference. Animals with a total exploration time less than 4 s were excluded from analysis. All behavioral experiments were performed in a randomized blinded setup.

Spatial working memory was assessed by using the Y-maze spontaneous alternation test, as previously described [41]. Each mouse was placed in the center of the Y-maze and was free to explore the arena for 6 min. The number of entries was counted per mouse: an entry required that both hind paws of the animal be placed completely inside the arm. A mouse would be making a triad when it visited all 3 arms consecutively. Between sessions, the maze was cleaned thoroughly with a 70% ethanol solution. As a measure for working memory, the percentage of correct alternations that the mouse made was calculated, being the number of triads divided by the maximum possible alternations (i.e., the total number of entries minus 2) × 100%. If a mouse scored significantly above 50% alternations (the chance level for choosing the unfamiliar arm), this was indicative of functional working memory.

### 2.9. Immunohistochemistry

For tissue staining, brain hemispheres were immersion-fixed in 4% paraformaldehyde (PFA) overnight at 4 °C, sunk in 15% to 30% sucrose-PBS, and embedded in OCT (Thermo Fisher). For this, 10 µm thick coronal cryosections were made using a Leica CM3050S cryostat and put on SuperFrost Plus slides (Thermo Fisher). 

Brain sections were stained for post-synaptic density protein 95 (PSD95) and the vesicular glutamate transporter 1 (vGlut1) to assess synaptic density in the hippocampus. Tissues were blocked for 1 h with 30% *v/v* donkey serum in Antibody Dilution Solution (AbDil: 150 mM NaCl, 50 mM tris-base, 1% *w/v* BSA, 100 mM L-lysine, 0.05% *v/v* Triton-X100, pH 7.4) at room temperature (RT), and then stained overnight with primary antibodies at 4 °C; mouse anti-vGlut1 (1/1000, MA5-27614 Thermo Fisher, Geel, Belgium); and rabbit anti-PSD95 (1/500, APZ-009 Alomone Labs, Jerusalem, Israel) in AbDil. After washing with PBS, slides were incubated for 1 h at RT with Alexa Fluor 488 Donkey-Anti-Rabbit IgG; (1:600; Thermo Fisher, Geel, Belgium A21206) and Alexa Fluor 555 Donkey-Anti-Mouse IgG (1:600; Invitrogen, Waltham, MA, USA, A31570) in AbDil with 5% *v/v* donkey serum. After washing with PBS, the bound secondary antibodies were stabilized by applying 2% *w/v* PFA for 10 min at RT. Samples were mounted using Fluoromount-G mounting medium (Thermo Fisher). Images were acquired on a Zeiss LSM900 using a 63× oil immersion objective (numeric aperture (NA) = 1.4) and the Airyscan detector. Per animal, 3 adjacent representative image stacks (resolution 1810 × 1810) from the same area of interest per slice were taken from 3 serial slices, as previously described [42]. Pre- and post-synaptic proteins were colocalized using Fiji ImageJ with the cut-off value to determine colocalization set at 4 pixels [43].

To stain for Aβ and assess plaque load, endogenous peroxidases were blocked by incubation with 3% H_2_O_2_ in methanol at RT. Slides were washed three times in 1× TBS 0.3% Triton X-100 and blocked in wash buffer supplemented with 5% bovine serum albumin (Sigma, A7906) and 1% goat serum for 1 h at RT. Primary human IgG Aβ antibody (clone 3D6), kindly provided by Dr. Mario Losen, was applied in a 1:8000 dilution and incubated overnight at 4 °C. After washing, slides were incubated with biotinylated secondary antibody (goat anti-human IgG antibody [H + L]), Biotinylated (BA-3000-1.5, Vector Laboratories, Newark, NJ, USA) for 30 min at RT. A three-part washing step was performed and diaminobenzidine (DAB, Dako, K3468) was applied for 1 min to develop the slides. Slides were washed three times and counterstained with hematoxylin (Leica, 3801582E) for 1 min. Cryosections were rinsed under running tap water and dehydrated in 70% (2 min) and 100% ethanol (2 min). Slides were cleared in xylene for 1 min and mounted using DPX (Merck) in a 1/10 dilution in xylene. Digital images were obtained using an Axio Scan Z1 Slide Scanner with 20× objective (N.A. = 0.8) (Zeiss, Zaventem, Belgium). The amyloid plaque load was quantified by using the analyze particles function in Fiji ImageJ.

### 2.10. ELISA of Distinct Pools of Aβ

Serial extraction of distinct pools of Aβ in hippocampal tissue was performed by employing a 4-step serial extraction by sonication and centrifugation, as previously described [44]. The tissue was first mechanically dissociated using the TissueRuptor (Qiagen) in 1 mL/150 mg wet weight TRIS buffer, pH 7.2 (50 mM TRIS, 200 mM sodium chloride, 2 mM EDTA, and complete protease inhibitors), with 2% protease-free bovine serum albumin. An aliquot of 10 µL was taken for measurement of total protein content. After centrifugation (21,000× *g*, 4 °C, 10 min), the supernatant was retained as the TRIS-soluble fraction. The pellet was homogenized with TRIS extraction buffer that contained 0.1% Triton X-100 and spun (21,000× *g*, 4 °C, 10 min), and the supernatant was retained as the Triton soluble fraction. The remaining pellet was homogenized in 2% sodium dodecyl sulfate (SDS) and spun (21,000× *g*, 4 °C, 10 min), and the supernatant was saved as the SDS-soluble fraction. The remaining pellet was homogenized in 70% formic acid (FA) and recentrifuged (44,000× *g*, 4 °C, 10 min), and the resulting FA-extracted supernatant was neutralized with 1M TRIS buffer (pH 11.0), representing the FA-extracted fraction. The total protein concentration (mg/mL) was determined for the tissue homogenate using the Pierce BCA Protein Assay Kit (Thermo Scientific) according to the manufacturer’s protocol and measured at 570 nm using the Clariostar Plus plate reader (Isogen Life Science, Utrecht, The Netherlands). A human Aβ42 sandwich ELISA (Invitrogen product number: KHB3441) was used to determine the Aβ42 concentration (pg/mg protein) of the third and fourth fraction according to the manufacturer’s protocol. The absorbance (at 450 nm) was read using the Clariostar Plus plate reader. Values were normalized for starting tissue wet weight and total protein content.

### 2.11. Statistical Analysis

GraphPad Prism 10.2.0 software (GraphPad software Inc., Boston, MA, USA) was used to perform statistical analyses. The sample size of each experiment was determined using G*Power-based power analysis. Outlier values were determined based on the Dixon test for extreme values (significance level of 0.05) and excluded from further analysis. Behavioral experiments were evaluated for differences compared to chance level using a one-sample t-test. Differences between groups were evaluated using a non-parametric Kruskal–Wallis test with Dunn’s post hoc analysis against the vehicle group when the sample size was *n* ≤ 5 or data were not normally distributed. When the sample size was *n* ≥ 6, normality was checked using the Shapiro–Wilk test. 

Normally distributed data were subsequently analyzed with a one-way ANOVA with Tukey’s test (in vivo and *post-mortem* analysis), Kruskal–Wallis with Dunn’s multiple comparisons test (colocalization analysis), two-way ANOVA (phagocytosis assay), and two-way ANOVA with Šídák’s multiple comparisons test (ELISA and surveillance treatment comparison). Differences in the relative area scanned per treatment for surveillance were analyzed by means of Brown–Forsythe and Welch ANOVA with Dunnett’s T3 multiple comparisons test. Immunohistochemical analysis of Aβ in vehicle- vs. A33-treated APPswe/PS1dE9 mice was analyzed by means of unpaired *t*-test. All data are displayed as mean ± standard error of mean (SEM), * *p* ≤ 0.05, ** *p* ≤ 0.01, *** *p* ≤ 0.001, **** *p* ≤ 0.0001 # *p* ≤ 0.05, ## *p* ≤ 0.01.

## 3. Results

### 3.1. Long-Term Inhibition of PDE4B Improves Cognition in 8-Month-Old APPswe/PS1dE9 Mice

Synapse loss has been shown by Hong and colleagues to be significantly higher in APPswe/PSEN1dE9 when compared with non-transgenic mice. APPswe/PSEN1dE9 mice had approximately 50 percent fewer puncta co-labeled for pre- and post-synaptic markers [45]. As PDE4B inhibition has been shown to be beneficial in alleviating neurodegeneration in AD, we propose that preemptive PDE4B inhibition will suppress the observed high levels of synapse loss seen in APPswe/PS1dE9 mice. 

Female monoallelic APPswe/PS1dE9 mice and WT littermates were injected daily with a PDE4B inhibitor, A33 (3 mg/kg), from postnatal 1–4 months, which is the time window that high levels of synapse loss are observed but before cognitive decline is to be expected (7–8 months) [45]. To assess whether A33 treatment slows or halts the progression of the AD state, animals were subjected to behavioral tests at 8 months of age, when cognitive decline has been described to appear in the APPswe/PS1dE9 model. The three groups were subjected to the OLT and Y-maze spontaneous alternation task to test spatial long-term and working memory, respectively. Four months after the end of daily treatment with A33 or vehicle, at the age of 8 months, cognitive decline was witnessed in untreated APPswe/PS1dE9 mice animals, as evidenced by deteriorating scores in both the OLT (Figure 2A) and Y-maze spontaneous alternation tests (Figure 2B) compared to WT littermates. Notably, these deteriorated scores were abolished in the A33-treated APPswe/PS1dE9 mice, as their scores were significantly different from 0 (in the OLT) and 50% (in the spontaneous alternation Y-maze test). 

### 3.2. Sustained Inhibition of PDE4B, Using A33, Does Not Alter Synapse Density of 4-Months-Old APPswe/PS1dE9 Mice but Normalizes Synapse Density in 8-Months-Old APPswe/PS1dE9 Mice

Both immediately after this therapeutic window at 4 months of age (t1) and after behavioral testing at 7 months of age (t2), synapse density was measured and analyzed through immunohistochemistry. Previous studies described that the earliest and most influential changes in synaptic elimination occurred in excitatory neurons; hence, we measured synaptic puncta in brain sections by immunocytochemistry with antibodies to vGlut1 and PSD95 [46,47]. Immediately after sustained long-term treatment with A33 (t1, 4 months-of-age), APPswe/PS1dE9 mice showed similar synaptic density (Figure 3A). Vehicle-treated APPswe/PS1dE9 mice that were kept for 3 additional months and underwent behavioral testing prior to sacrifice (t2, 8 months of age) showed a significant decrease in synaptic density when compared to WT littermates, an effect that was attenuated by treatment with A33 (Figure 3B). Staining of vGlut1 and PSD95 is visualized in Figure 3C (representative pictures of every group in Appendix A). The number of colocalizations was quantified as a measure of the number of synapses per field of view (78.01 µm × 78.01 µm). 

### 3.3. Inhibiting PDE4B Using A33 Reduces SDS-Soluble Aβ42 Levels but Does Not Affect Plaque Load

In addition to synapse loss, the accumulation of Aβ and plaque formation is noteworthy in the progression of AD. Microglia attempt to clear neurotoxic amyloid depositions in order to counteract the buildup and formation of plaques; however, this has been proposed induce them to become reactive and pro-inflammatory, potentially causing collateral damage to nearby synapses [48]. To test if inhibition of PDE4B has effects on clearance of amyloid, the sodium dodecyl sulfate (SDS)- and FA-soluble (formic acid solvent) fractions were tested for Aβ content by means of an ELISA, as these fractions have been shown to correlate with cognitive decline [44]. Detection of Aβ42 in distinct fractions of the brain tissue (Figure 4A) revealed an overall decrease in amyloid in A33-treated mice when compared to vehicle-treated APPswe/PS1dE9 mice (*p* = 0.0167, two-way ANOVA). This difference was reflected in the SDS-soluble fraction (*p* = 0.0445) but not in the FA-soluble fraction (*p* = 0.4412). Animals from the WT group were not included in the analysis as these animals do not express human Aβ [49]. Immunohistochemistry revealed no changes in plaque load (Figure 4B).

To attempt to isolate the microglial effect, an additional experiment was performed that employs bone marrow transplantation of *Pde4b^+/+^*, *Pde4b^+/−^,* and *Pde4b^−/−^* cells to replace the resident microglia in lethally irradiated APPswe/PS1dE9 mice, starting at 6 weeks of age to assess the same time window as the treatment did in vivo using injections. However, due to unforeseen mortality, the experimental setup encountered a power issue (e.g., *n* = 2 per group) to further study behavior and perform *post*-*mortem* brain analyses. A repeat of the experiment was performed to validate radiation-induced cardiac damage as the cause, detailed in the Appendix A.

### 3.4. Inhibition of PDE4B Does Not Lower Microglial Phagocytic Capacity of Synaptosomes In Vitro

To test the effect of PDE4B on phagocytosis, WT microglia were pre-treated with the PDE4B inhibitor A33 for 1 h and exposed to pHrodo-labeled synaptosomes for 40 min. The particles are non-fluorescent outside the cell but fluoresce brightly red when taken up in the phagosomes due to acidification of the pH. These measurements showed an equal number of cells that had phagocytosed synaptic material (Figure 5A). No difference in the fluorescence intensity per cell was seen for vehicle- and A33-pretreated cells (Figure 5B). Despite no significant differences in fluorescence intensity per cell between vehicle- and A33-pretreated cells, the intensities did seem consistently lower in A33-pretreated cells. Unpaired one-tailed t-test of the area under curve (AUC) was significantly lower in A33-treated cells when compared to vehicle-treated cells (*p* = 0.0281). Furthermore, *Pde4b^+/+^*, *Pde4b^+/−^,* and *Pde4b^−/−^* microglia were tested for phagocytic capacity, and *Pde4b^−/−^* microglia had a consistent significantly higher phagocytic capacity compared to microglia from *Pde4b^+/+^* littermates. Additionally, *Pde4b^+/−^* had significantly higher phagocytic capacity compared to microglia from *Pde4b^+/+^* littermates at two timepoints: 2–3 h and 23.5 h. However, the graph here also seemed consistently higher for *Pde4b^+/−^* miroglia. One-way ANOVA of the AUC was significantly higher in both *Pde4b^+/−^* and *Pde4b^−/−^* microglia when compared to *Pde4b^+/+^* cells (*p* < 0.0001). More recent research has found discrepancies between knockout and knockdown approaches when studying genetic loss of function. An explanation for this is genetic compensation [50]. To examine if there is compensation of other PDEs on cAMP, the expression pattern of *Pde4a*, *Pde7a*, *Pde7b*, and *Pde8a* was examined, shown in Figure 4. Conversely, the *Pde4a* gene expression closely followed that of *Pde4b*. *Pde7a*, on the other hand, had normal expression in the *Pde4b*^+/−^ microglia, while its expression was significantly increased in the *Pde4b*^−/−^ microglia. 

### 3.5. Microglia Increase Surveillance of Their Micro-Environment upon PDE4B Inhibition In Vitro

Microglial surveillance has been shown to be cAMP-dependent, which can be modulated by inhibition of PDE3B or non-specific inhibition using IBMX [20]. To investigate the beneficial effect of PDE4B inhibition on microglial homeostatic behavior, we aimed to investigate the effect of this specific cAMP-signaling nanodomain [51,52,53,54]. The administration of the PDE4B inhibitor A33 causes microglia to increase their surveillance, indicated by increased area scanned (µm^2^), which is measured as the surface area of filopodia (Figure 6). The experiment took advantage of the ubiquitous expression of eGFP in the Cx3cr1^eGFP/-^Cx3cr1^eGFP/+^ mice to visualize the highly dynamic filopodia [55]. Increased surveillance indicates a more homeostatic phenotype for microglia, possibly leading to less targeting and degrading of synapses [56].

## 4. Discussion

Multiple studies have highlighted synaptic elimination and subsequent neuronal loss as crucial, early contributing factors to the pathogenesis of AD [57,58,59,60]. Microglia are active players in the maintenance of synaptic density. However, early treatment to prevent synapse loss in the context of AD has not been extensively studied and it is not known whether prevention of synapse loss can battle cognitive decline in AD. This study aimed to examine the effects of PDE4B inhibition with the goal to prevent microglia targeting synapses to retain synaptic density.

Microglia have been shown to influence synapse number and affect cognition in AD [45]. PDE4B is a major regulator of microglial functioning, prompting the hypothesis that inhibiting PDE4B in microglia, when excessive synapse loss is expected to occur, could have beneficial effects. To test this hypothesis, APPswe/PS1dE9 mice and WT littermates were injected daily with a PDE4B inhibitor, A33 (3 mg/kg), and subjected to the OLT and Y-maze spontaneous alternation task at 8 months to test spatial long-term and working memory, respectively. These cognitive capacities were restored to levels on par with WT animals by A33 treatment. This is consistent with other PDE4 inhibitor studies, such as with roflumilast, which have previously been shown to improve cognition acutely but not pre-emptively in both rodents and humans. Specifically, by administering roflumilast to mice three hours after the first learning trial (during consolidation) and to humans one hour before testing (to affect memory acquisition), cognitive capacities are improved.

*Post-mortem* analyses of animals in the present study were consistent with the differences seen in cognitive capacities. The synapse number of animals at t2, after being subjected to behavioral testing, showed significant improvement in comparison to untreated APPswe/PS1dE9. It is unclear if this effect is mediated solely by microglia, and what happens to the microglia and synaptic density when long-term inhibition of PDE4B has ceased is unknown. Possibly, the treatment does not alleviate excessive synapse loss but rather creates an environment that allows activity-dependent synaptogenesis or synaptic plasticity. When synapses are actively used, in memory signaling for instance, connections are kept and strengthened by the presence of “stabilization” signals [61,62]. Moreover, microglia play an active role in learning-dependent synapse formation through brain-derived neurotrophic factor (BDNF) signaling [63]. Long-term inhibition of PDE4B spanning the time window that has been verified to heavily feature synaptic loss has positive effects on cognitive decline and synapse loss later in life [64]. As evidenced by the in vitro experiments, surveillance is increased and phagocytosis could possibly be affected by inhibition of PDE4B. It would be of interest to research whether these effects are reproduced in vivo. Potentially, it could be possible that the microglia positively affect synaptic plasticity, although our experimental setup does not definitively prove this. As a direct consequence of increased synaptic plasticity, the synaptic connections are strengthened, leading to less weak or surplus synapses that are flagged for removal and subsequently phagocytosed. If further research would uncover these effects in vivo, it would be in line with our data, showing higher synapse density despite phagocytic capacity being unaltered.

In the SDS-soluble fraction, which is believed to represent the membrane-associated Aβ, we detected less Aβ in the A33-treated as opposed to vehicle-treated mice. Surprisingly, this effect was not reflected in the FA-soluble fraction, which is believed to represent the fraction that is insoluble in the brain. In a study performed on human AD patients, faster cognitive decline was associated with elevated temporal SDS-soluble Aβ_42_ levels, whereas slower decline was associated with elevated cingulate FA-soluble and SDS-soluble Aβ42 levels [44]. Previous work showed no effect on Aβ content in these fractions after long-term exposure to rolipram, an unspecific PDE4 inhibitor [65]. Possibly, the inhibition of PDE4B does have some effect on the clearance of SDS-soluble Aβ42 mediated by microglia, but the targeting of multiple genes by rolipram blocks this effect. Further validation, by means of immunohistochemical analysis, showed that plaque load was similar in vehicle-treated APPswe/PS1dE9 animals as compared to those treated with A33. This is in line with the findings of the ELISA, showing that A33 does not influence the FA-soluble fraction. Similarly, Sierksma and colleagues reported that inhibition of PDE4D by means of Gebr-7b improves cognition but does not affect plaque formation in a treatment window of 3 weeks starting at 19 weeks of age [66]. Based on these findings, we can conclude that long-term PDE4B inhibition positively influences synapse number. Additionally, this treatment seemed to prevent cognitive decline. These effects seem to happen independently from Aβ deposition, although there was a marked decline in SDS-soluble Aβ42. Aβ has been shown to activate microglia: PDE4B is shown to be highly upregulated when microglia are exposed to both the soluble and the fibrillary forms of Aβ. This is accompanied by upregulation of pro-inflammatory molecules such as tumor necrosis factor-alpha (TNFα), interleukin-1beta (IL-1β), and macrophage inflammatory protein (MIP-1α). Interestingly, inhibiting PDE4 with rolipram (0.1 µM) has been shown to counteract the upregulation of TNFα [67].

To investigate the cellular mechanisms behind these findings, in vitro analyses were performed to identify the effects of PDE4B inhibition on microglial functioning. Two major mechanisms of action that are necessary for the detection and clearance of synapses are surveillance and phagocytosis. Engulfment and elimination of synapses by microglia are key to creating and maintaining a well-balanced neuronal circuitry. In particular, synapse engulfment and phagocytosis were shown to be directed by the complement system [45,68]. C1q and C3 have been shown to mediate targeted microglial phagocytosis of hippocampal synapses both in healthy individuals and AD patients. Microglia contact and/or engulf synaptic elements in an activity-dependent manner, mediated by the fractalkine receptor Cx3Cr1 [69]. It has already been shown that macrophage phagocytosis is regulated by cAMP, yet the exact effect differs greatly depending on the target of phagocytosis and the associated mechanism of action. For instance, elevation of cAMP using cell-permeable analogues of cAMP or by receptor-directed stimuli inhibits macrophage phagocytosis of apoptotic cells as well as bacteria or other debris [21,23,70]. Mechanistically, increased cAMP levels reduced expression of complement receptors via protein kinase A (PKA) and exchange proteins activated directly by cyclic AMP (Epac) pathways [23,71]. However, the administration of A33 in the present study showed little to no effects on microglial phagocytic capacity on the level of whole synaptosomes. More specifically, neither the total number of WT microglia that phagocytosed nor the amount of synaptosomes phagocytosed per cell were significantly lowered by A33 administration. However, when analyzing the AUC of both graphs, an unpaired one-tailed t-test revealed a significantly lower AUC in the A33-treated cells when compared to the vehicle-treated cells (*p* = 0.0281). It has been shown before that there is a dual role for cAMP and PKA in phagocytosis: activation at normal cAMP levels and inhibition at higher. There is a narrow physiological window of cAMP levels that ensures efficient downstream effector mechanisms. This window is tightly balanced by cAMP production by adenylate cyclase and cAMP degradation by PDEs, which maintains normal operating cAMP levels that enable efficient phagocytosis. It is possible that pre-treatment with A33 for one hour does not sufficiently alter the cAMP levels in the right nanodomain for effects to be witnessed in this setup. More specifically, cytoskeletal changes leading to filopodia formation do seem to be targeted, while formation of phagocytic cups might be regulated via a different signaling domain [53,54,72,73]. 

As an additional readout for microglial PDE4B involvement in phagocytosis, the phagocytotic capacity of pHrodo-labeled synaptosomes by *Pde4b^+/+^*, *Pde4b^+/−^*, and *Pde4b^−/−^* murine microglia was investigated to elucidate the role of *Pde4b* on microglia-mediated synaptic elimination in vitro. Surprisingly, microglia with *Pde4b^+/−^* expression had more phagocytosing cells compared to *Pde4b^+/+^* microglia, while *Pde4b^−/−^* microglia only demonstrated an augmented number of phagocytosing cells than *Pde4b^+/+^* microglia during the peak time of phagocytosis. Moreover, *Pde4b^+/−^* microglia had increased phagocytotic capacity compared to *Pde4b^+/+^* and *Pde4b^−/−^* microglia. PDE4B is the predominant PDE4 subtype expressed by microglia. Heterozygous or no expression of *Pde4b* should alter microglial cAMP levels. However, the heterozygous expression or complete absence of *Pde4b* can potentially be compensated by modified gene expression of other *Pde* genes. Indeed, a significant upregulation of *Pde7a* was observed in our data of unstimulated *Pde4b^−/−^* murine microglia. *Pde7* expression is low under physiological conditions in microglia [74]. Nevertheless, *Pde7* inhibition in vivo reduced microgliosis and pro-inflammatory cytokine production in an experimental autoimmune encephalomyelitis model and spinal cord injury in mice, respectively, which indicates that PDE7 can influence microglia activity [75,76]. As a consequence, it is hypothesized that a physiological elevation of cAMP is present in *Pde4b^−/−^* microglia due to the upregulation of *Pde7a*. *Pde4b^+/−^* did not show altered cAMP-specific *Pde* expression except for lower *Pde8a* expression. Other cAMP-specific or dual-specific *Pde* subtypes can be measured in the future to have a comprehensive knowledge of feedback mechanisms. It would be highly interesting to identify cAMP levels in distinct nanodomains in both *Pde4b* genotypes to assess compensation on a functional level.

Microglial surveillance is the mechanism by which microglia actively scan their micro-environment. As active patrolling immune cells, microglia extend and retract short filopodia to sense not only brain damage but also synaptic activity [77,78,79]. Microglial surveillance is shown to be affected by cAMP levels [20], and lipopolysaccharide (LPS)-induced inflammation decreases basal surveillance of synapses [80]. Furthermore, it has already been shown that inhibition of the dual substrate hydrolyzing PDE3B, by means of cilostamide and general PDE inhibition using IBMX, influences the cAMP levels in microglia to alter filopodia surveillance [20]. PDE4B is highly present in microglia and regulates cAMP and inflammatory pathways, so it stands to reason that it exerts marked effects on surveillance. Indeed, the present study shows that PDE4B inhibition alters microglia surveillance: the surveilled area increased in every cell imaged upon PDE4B inhibition. This is reflected in the heat map images, quantifiably showing a marked increase in filopodia motility. Taken together, these results suggest that suppression of PDE4B functioning drives microglial activity to a more homeostatic phenotype. These alterations might potentially play a role in preventing microglia from adopting their ‘disease-associated microglia (DAM)’ phenotype in vivo, although this should be further investigated [81]. This would mean that phagocytosis is not directly altered, rather the microglial baseline homeostatic behavior is stimulated.

These findings demonstrate the potential of an early intervention using A33, but inhibition affects every cell type that expresses PDE4B, such as astrocytes and neurons [82,83]. Some caveats should be kept in mind when considering our used experimental setup. For instance, we did not include a WT group that was treated with A33. While adding this group would enable a stronger two-way ANOVA statistical analysis, we do not expect any improvement in treating a WT group since a ceiling of performance has been reached, as we saw previously when testing cognition enhancers in this mouse model. In line with this, previous research from our group has demonstrated that cognitive performances remain uninfluenced by pro-cognitive treatments in wild-type mice at the used ORT/OLT time interval [40]. Additionally, the role of microglial functioning is not thoroughly proven by our *post*-*mortem* experiments. Since PDE4B inhibition by means of administration of A33 is a non-selective form of treatment, the effects witnessed on synapse density could stem from a neuroprotective effect. The strengthening of synapses could possibly be attributed to the actions of A33 on neuronal or astrocytic functioning [84].

The effects of regulating PDE4B function on cognition and memory formation have previously been investigated, pointing towards facilitation of both memory acquisition and consolidation [85,86]. Furthermore, a recent publication of Armstrong and colleagues investigated the protective effect of PDE4B subtype-specific inhibition in an App knock-in mouse model. In line with our results, they also reported cognitive protective effects, associated with no decrease in amyloid plaque load [87]. Interestingly, they witness an improved cerebral hypometabolism. However, in contrast to Armstrong and colleagues, who made use of a hypomorphic mutation (Pde4b^Y358C^) that decreases PDE4B’s cAMP hydrolytic activity, our research highlights for the first time the relevance of pharmacological inhibition of PDE4B to reduce synapse loss and, therefore, explores a novel underlying mechanism of PDE4B inhibition. Furthermore, our study employs PDE4B inhibition only in a strict time window, whereas the model used by Armstrong and colleagues causes profound and long-lasting changes starting from birth and persisting throughout the entire lifespan. Even though memory formation occurs after cessation of treatment with A33, any possible auxiliary effects of PDE4B inhibition on other cell types cannot be excluded. As such, further research is warranted to investigate the role of microglia in the effects witnessed in vivo.

Microglia are involved in a broad array of functions and the pathology of AD is a long and dynamic process. Therefore, identifying and selectively targeting the exact nature of microglia dysfunction at the correct time remains an elusive feat. Treatment of animals before cognitive deficits appear has been conducted before, but not in the context of attempting to cease early synaptic pruning by modulating cAMP in microglia. For example, Gong and colleagues showed marked effects on synaptic and cognitive functions after a 3-week treatment period of rolipram in 3-month-old APPswe/PS1dE9 mice [65]. Similarly, 3-week-long inhibition of PDE4 by the prototype inhibitor FCPR03 counteracted the synapse loss caused by chronic unpredictable mild stress in adult mice [88]. Furthermore, it has been shown that PDE4 inhibition using GEBR-7b (0.001 mg/kg) treatment or PDE2 inhibition during behavioral testing in the time window of 5–8 months of age has a marked effect on cognition [41,66]. Finally, a recent paper showed, using a similar experimental setup to the present study, that therapeutic inhibition of microgliosis by means of minocycline is most effective against AD pathology (i.e., microgliosis, Aβ levels and cognitive decline), when started before the onset of microgliosis and provided continuously [89]. Notably, it has been shown that minocycline upregulates cyclic-AMP response element binding (CREB) and BDNF in the hippocampus of cerebral ischemia rats and improves behavioral deficits, suggesting a shared pathway between minocycline and cAMP signaling [90].

Multiple studies using PDE inhibition have been conducted focusing on the neuronal aspect of the disease, while there is a knowledge gap on microglial functioning in this context [33,91]. The current research partially addressed this notion and showed that PDE4B inhibition positively influences synapse number, accompanied with a small but significant change in Aβ levels. Importantly, the effects of A33 administration on neurons should not be disregarded, but the surveillance data support the idea that microglial functioning is altered and might play a role in the effects witnessed in vivo. However, the non-neuronal role of PDE4B inhibition should also be taken account, in addition to the anti-inflammatory effects of PDE4B inhibition. More specifically, PDE4B inhibition has been shown to reduce microglial activation through suppression of inflammatory factors and acts as an anti-inflammatory by decreasing inflammatory microglia activation [27,82]. Our experimental setup did not investigate a potential contribution of the reduced inflammatory state of microglia, so this effect on long-term cognitive performances cannot be excluded. Furthermore, the present study showed that the cognitive decline witnessed in AD may be partially slowed down using long-term A33 treatment in the correct time window. In doing so, this study proposes an innovative way of preventing cognitive decline, one of the major symptoms of AD. Future studies are needed to validate the therapeutic potential of inhibitors in a preventative manner, as well as targeting microglia directly.

## Figures and Tables

**Figure 1 cells-13-01000-f001:**
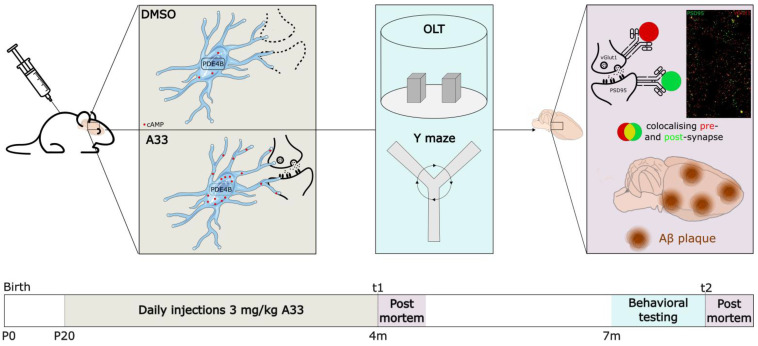
Experimental design. Schematic representation of the experimental design for the in vivo examination of PDE4B inhibition in APPswe/PS1dE9 and wild-type (WT) mice. Animals were injected daily with DMSO or A33, starting at postnatal day (P) 20, until P120. Immediately after the completion of the treatment, four animals per group were sacrificed for interim *post-mortem* analyses on synapse density (t1). The remaining animals (*n* ≥ 10) were housed for another 3 months without additional interventions. At the age of 7 months, behavioral tests were performed. Finally, all animals were sacrificed for *post-mortem* analyses on synapse density and amyloid β (Aβ) load (t2).

**Figure 2 cells-13-01000-f002:**
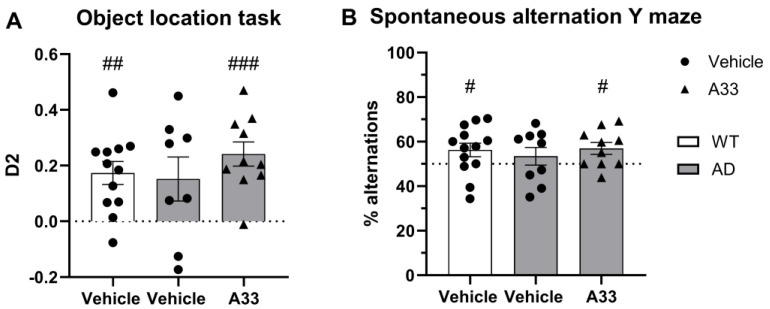
Preventative treatment with A33 counteracts cognitive decline significantly. (**A**) Spatial memory was evaluated by means of the object location test (OLT) at a 5 h inter-trial interval. Data are displayed as mean ± SEM (one-sample *t*-test [compared to 0]; ## *p* < 0.01; ### *p* < 0.005; *n* = 12, 8 and 10, respectively). D2 = discrimination index. (**B**) The Y-maze spontaneous alteration task was used to evaluate spatial working memory. Data are displayed as mean ± SEM (one-sample *t*-test [compared to 50%]; # *p* < 0.05; *n* = 13, 9, and 10, respectively).

**Figure 3 cells-13-01000-f003:**
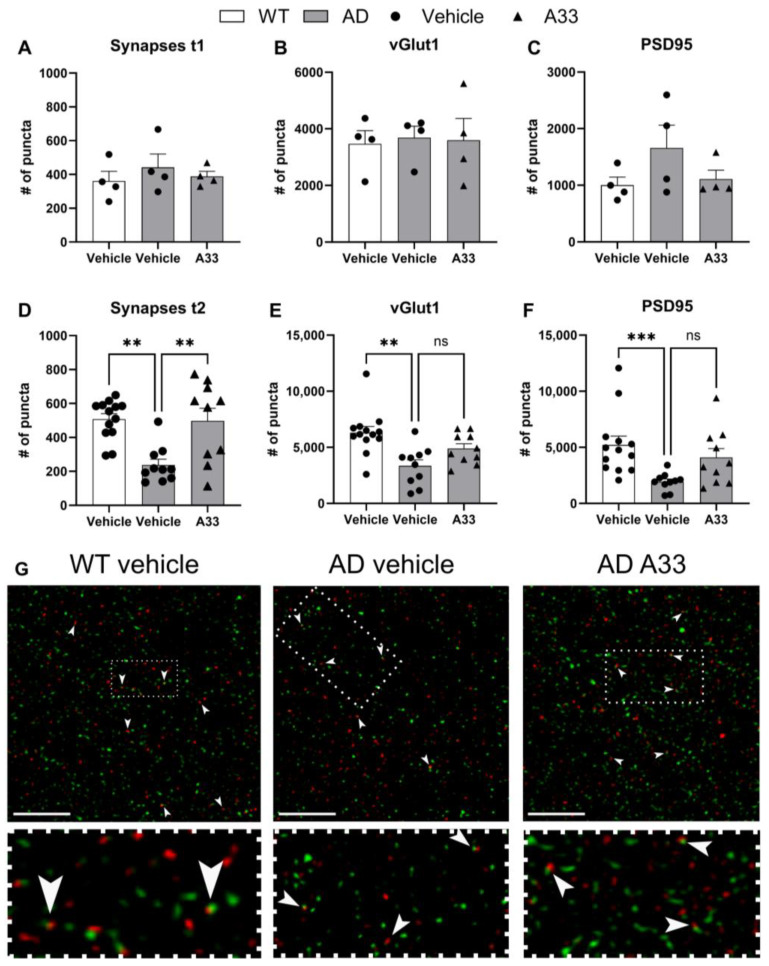
Synapse number seems unaffected immediately after sustained treatment, yet is significantly retained at 8 months of age. (**A**) The number of colocalizing pre- and post-synaptic proteins per group. (**B**) The number of puncta indicating pre-synaptic protein vesicular glutamate transporter 1 (vGlut1) per group (**C**). The number of puncta indicating post-synaptic density protein 95 (PSD95) per group. (**A**–**C**) represent data from t1 (4 months of age), *n* = 4, visualized as mean ± SEM (analyzed by means of Kruskal–Wallis with Dunn’s multiple comparisons test). (**D**) The number of colocalizing pre- and post-synaptic proteins per group. (**E**) The number of puncta indicating pre-synaptic protein vGlut1 per group. (**F**) The number of puncta indicating post-synaptic protein PSD95 per group. (**D**–**F**) represent data from t2 (8 months of age), (*n* ≥ 10), visualized as mean ± SEM (analyzed by means of Kruskal–Wallis with Dunn’s multiple comparisons test; ** *p* < 0.01, *** *p* ≤ 0.001, ns = not significant). (**G**) Representative figure of the synapse staining for animals at 8 months of age, showing the pre-synaptic protein vGlut1 in red and PSD95 in green. White arrows indicate sites of colocalization (yellow) and white scale bar represents length of 1 µm. White dotted line indicates region that is further zoomed into below the pictures.

**Figure 4 cells-13-01000-f004:**
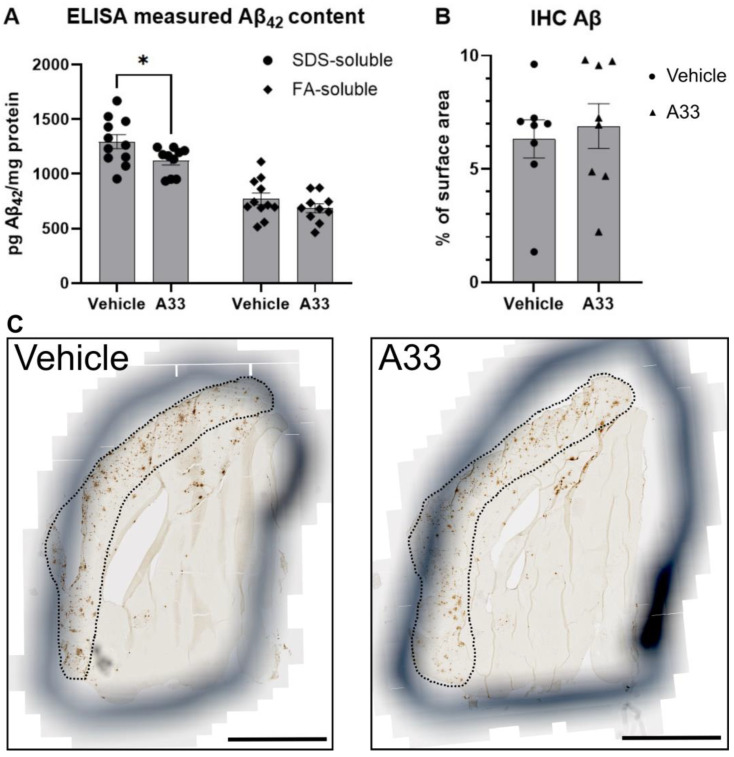
Treatment with A33 leads to a decrease in soluble and overall amyloid β (Aβ42) but not in FA-soluble Aβ42 and plaque formation. (**A**) Distinct Aβ42 forms measured by means of ELISA assessing sodium dodecyl sulfate (SDS)- and formic acid (FA)-soluble fractions of Aβ42 in APPswe/PS1dE9 animals. Data are represented as mean ± SEM (two-way ANOVA with Šídák’s multiple comparisons test; * *p* < 0.05). (**B**) Aβ plaque load as quantified at 20× magnification in the brain of APPswe/PS1dE9 mice by immunohistochemical staining of Aβ (*n* = 8). Plaque load was calculated as the percentage of Aβ positive area in the total surface area. Data are represented as mean ± SEM (analyzed by means of Mann–Whitney test). (**C**) Representative pictures of sectioned tissue from one murine hemisphere taken at bregma anteroposterior −0.5 mm, immunohistochemically stained for Aβ plaques and imaged by means of Axio Scan Z1 slide scanner. Area defined by dotted line is cortex. Black scale bars represent length of 1000 µm.

**Figure 5 cells-13-01000-f005:**
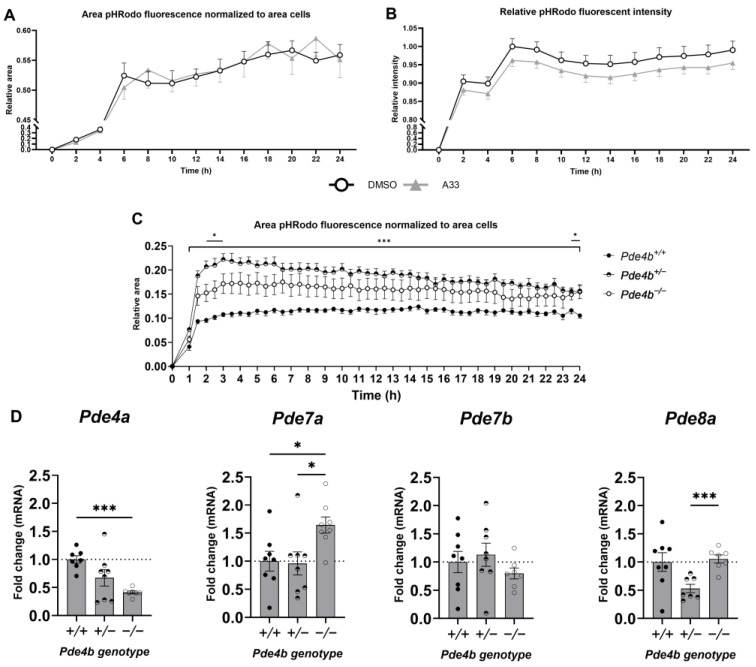
Microglial phagocytosis of pHrodo-labeled synaptosomes is unaltered when pre-treated with the PDE4B inhibitor A33 for 1 h, but *Pde4b^+/−^* and *Pde4b^−/−^* microglia’s phagocytic capacity is increased, possibly due to compensation by other cAMP-hydrolyzing PDEs. (**A**) Area of red fluorescent pHrodo normalized for total cell area. (**B**) Intensity of red fluorescent pHrodo normalized for total cell area. Microglia isolated from postnatal day (P)0–3 WT pups were pre-treated with 100 nM A33 for 1 h and subsequently exposed to pHrodo-labeled wild-type synaptosomes for 40 min. Pictures were taken at 2 h intervals using IncuCyte (Sartorius) and both the area and intensity of red fluorescence were normalized for total cell area (*n* = 8, 250,000 cells/well). (**C**) Area of red fluorescent pHrodo normalized for total cell area of *Pde4b^+/+^*, *Pde4b^+/−^,* and *Pde4b^−/−^* microglia. Microglia isolated from postnatal day (P)0–3 PDE4B KO pups were pre-treated with 100 nM A33 for 1 h and subsequently exposed to pHrodo-labeled wild-type synaptosomes for 40 min. Pictures were taken at 0,5 h intervals using IncuCyte (Sartorius) and the area of red fluorescence was normalized for total cell area (*n* ≤ 8, 250,000 cells/well). Data are represented as mean ± SEM and analyzed by means of two-way ANOVA or mixed effects analysis, * *p* < 0.05, *** *p* < 0.001. (**C**,**D**) mRNA expression of cAMP-specific phosphodiesterase Pde4a, Pde7a, Pde7b, and Pde8a expressed relative to Pde4b^+/+^ expression of respective genes normalized with Cyca and Ywhaz as reference genes. Data are represented as mean ± SEM and analyzed with a Welsh ANOVA test with Dunnets’s correction for multiple comparisons * *p* < 0.05, *** *p* < 0.001, *n* = 6–8 biological replicates.

**Figure 6 cells-13-01000-f006:**
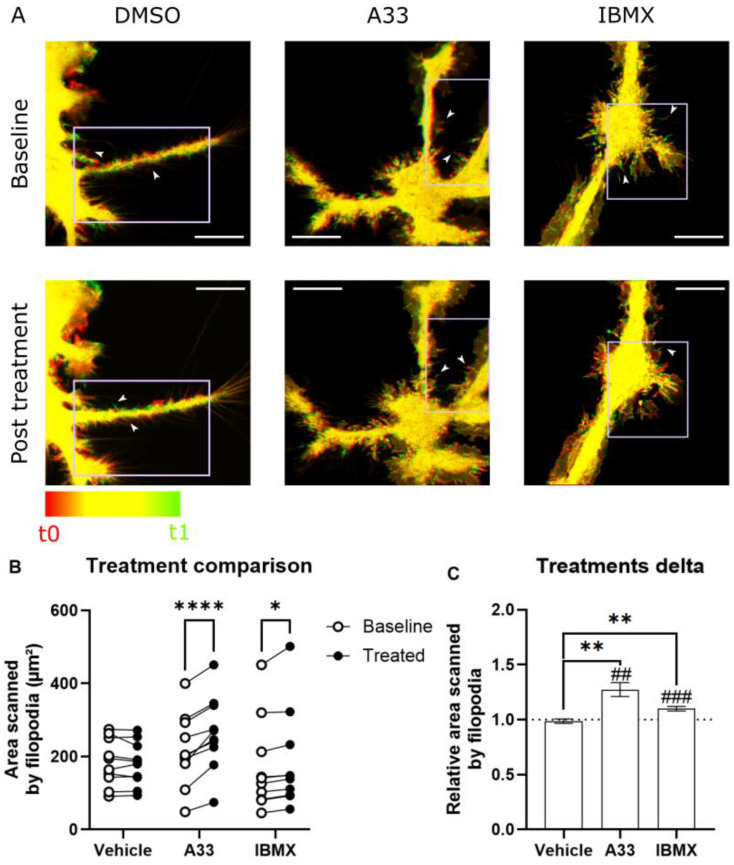
Microglial filopodia are regulated by PDE4B. (**A**) Representative Airyscan images of selected regions in untreated, A33- (100 nM), and IBMX- (100 µM) treated microglia. Visualized are temporal color-coded overlaid images taken 2.5 min apart, where red and green represent the initial and final area occupied by the cell, respectively. Yellow indicates no movement. The color-coded bar indicates the time range and magenta boxes indicate regions of interest. (**B**) Total scanned area by filopodia in selected regions (*n* = 10 independent cells) analyzed by means of two-way repeated measures ANOVA with Šídák’s multiple comparisons test. * *p* < 0.05, **** *p* < 0.0001. (**C**) Mean area scanned by filopodia, relative to baseline numbers, in selected regions per treatment. Data are represented as mean ± SEM. Analyzed by means of Brown–Forsythe and Welch ANOVA with Dunnett’s T3 multiple comparisons test. ** *p* < 0.01, ## *p* < 0.01, ### *p* < 0.001. White arrows indicate filopodia and white scale bar represents length of 10 µm.

**Table 1 cells-13-01000-t001:** Used primers for qPCR analysis.

*Gene*	Forward Primer	Reverse Primer
*Pde4a*	GCCTTGCACTGAGGAAACTC	GGCTGTCTCCTGCTTCAAAC
*Pde7a*	TGGAGGCTCAGATAGGTGCT	CCAGTTCCGACATGGGTTAC
*Pde7b*	ATCGCTTGACAAATGGGAAC	GGGTTGTGACCGTGGTAATC
*Pde8a*	TGGCTGTGCTCTACAACGAC	CCGGTAGTCATTCCTCTCCA
*Cypa*	GCCTCTCCTTCGAGCTGTT	AAGTCACCACCCTGGCA
*Ywhaz*	AGCCGAGCTGTCTAACCGAG	GCCAACTAGCGGTAGTAGTCA

## Data Availability

The data that have been used are confidential.

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
