# Peer review of "Early Inhibition of Phosphodiesterase 4B (PDE4B) Instills Cognitive Resilience in APPswe/PS1dE9 Mice"

_cells, 2024, doi:10.3390/cells13121000_

Round 1

Reviewer 1 Report

Comments and Suggestions for Authors

The authors conducted a thorough experiment to investigate the effects of PDE4B inhibition with A33 on synaptic loss, spatial long term & working memory, amyloid b level, phagocytosis & surveillance through microglial action.

 Comments:

1.   If mice were tested for OLT & Y-maze when A33 treatment ended at t1 (4 month age), together with testing results at t2 shown in Fig. 2, it will be a nice kinetic results and support the findings for Fig.3A & 3B.

2.   Fig. 2 legend showed ** but the graphs have ##.

3.  Fig. 2A & 2B: the significance bar should cover the groups compared for easy viewing (as shown in Fig. 3B).

4.  Fig. 3C: Are these staining from 4 months or 8 months animals?   AD vehicles seemed to have less green (PSD95) than the other 2 groups.

5.    In Methods Section under BMT:  Mice at 10-wk-old were irradiated & immediately followed by BMT.  One week later PLX5622 was given for 2 wks.

a)   Will PLX5622 impair microglia-like cells in BMT? 

b)   Appswe/PS1dE9 mice at 10-wk-old have synaptic loss so their baseline would be different from WT before BMT.

c)    If without irradiation (immune cells will not be affected), using only PLX5622 to ablate microglia for X weeks followed by BMT, will microglia-like cells repopulate CNS to achieve the goal?

6.  In Fig. 5B: although the fluorescence intensity per cell was no difference in vehicle - & A33-pretreated cells, the graph did show a consistent lower intensity in A33-pretreated cells.

7.    In Fig. 5: The WT p0-3 microglia were used.  If Appswe/PS1dE9 p0-3 microglia were used, perhaps the results would be more relevant.

8.   In Fig. 6: MACS method was used to isolated microglia from Cx3CR1eGFP/+ mice (p21).  Why not use p0-3 postnatal Cx3CR1eGFP/+ pups with the shake-off method as in WT?  Microglia isolated with MACS method showed different responses when exposed to inflammatory cytokines.

Reviewer 2 Report

Comments and Suggestions for Authors

This manuscript presents the effects of PDE4B-specific inhibitor A33 on the mouse model of Alzheimer’s disease.  The authors examined changes in spatial long-term and working memory, synapse density, and amounts of amyloid beta and plaque formation. Using isolated microglia, they examined the effects of A33 on phagocytotic activity and surveillance activity. Although preliminary, they also examined the effects of bone marrow transplantation of different genotypes of Pde4b (+/+, +/-, or -/-). The authors try to stress the role of microglia in the A33-mediated improvement of AD symptoms. However, the evidence presented does not warrant the direct involvement of microglia. They should tone down a bit. For example, the current title is a bit too strong. The manuscript is well-prepared overall, but there are some typos.  Some descriptions are inaccurate and difficult to understand, especially in the section on the cardiac health of BMT animals. This section should be revised. The following are specific points to be considered.

L116. The genotype of Pde4b has a typo. There are several incidences of Pde4B in other parts of the manuscript. They should be fixed.

L188. 30,000, not 30.000.

L376. The figure shows #, not *. They should be fixed.

L379. “n = 13, 3 and 10 respectively” Is 3 a typo?

Figure 4C. Need the description of the dotted line enclosed area.

L437-438. “using injection” Be more specific.

L447. S1, not S2

L461. S2, not S3

L461-462. This is not accurate.

L466-467. This is not accurate.

L454-472. Overall, very difficult to understand which animal groups are compared in the text.

Reviewer 3 Report

Comments and Suggestions for Authors

In the present study the Authors explore the role of Phosphodiesterase 4B (PDE4B) in cognitive decline and progression of disease in an experimental model of Alzheimer’s diseases, represented by APPswe/PS1dE9 mice. They inhibited PDE4B by A33 (3 mg/kg/day) administration in a time frame ranging between 20 days-of-age to 4 months-of-age and then evaluated cognitive function (object location task and Y-maze) at 7-8 months of age. Synaptic loss was quantitatively assessed in hippocampal samples both at the end of A33 treatment and after cognitive evaluation, at 7-8 months of age. They found that A33 treatment improves cognitive decline, reduces synaptic loss and decreases soluble and overall amyloid β (Aβ42), but not FA-soluble Aβ42 and plaque formation in APPswe/PS1dE9 mice. Moreover, they hypothesize a possible role of microglia. Therefore, in vitro experiments were performed, inhibiting PDE4B in primary microglial cultures. The treatment induced an increase of microglial filopodia formation, suggestive of the acquisition of a surveillant phenotype. Their conclusion is that that regulation of PDE4B activity could prevent microglia from adopting the disease-associated microglia phenotype in vivo.

Although of potential interest and well written, the topic is not new and the role of phosphodierase inhibition in APPswe/PS1dE9 mice on cognitive decline and synaptic loss had been explored also in other studies.

The hypothesis that microglia play a relevant role in the protective effect exerted by PDE4B inhibition is interesting, but the results obtained by the in vitro experiments are scarce. For instance, no evidence of a reduced response to inflammatory stimuli with consequent reduced production of inflammatory cytokines after PDE4B inhibition has been provided. Therefore the conclusions seem to overstate the results.

In addition, there are some points the need to be assessed. In particular:

A control group of wt A33-treated animals should have been performed. Therefore, my suggestion is to add this group and perform a two way Anova statistical analysis.

Fig. 3C is not clear and should be improved. Double staining is barely detectable. The image must be replaced with another at higher magnification counterstained -for instance- with  a fluorescent dye such as Neurotrace.

Similarly, Fig 4c is unclear and of poor quality. The anatomical region is not detectable and plaques are invaluable.

Results related to the in vivo experiments on microglia depletion should be deleted from the main text. The experiment, for several reasons, in actual facts, was not performed.

Further experiments to clarify the effects of PDE4B inhibition on microglia should be performed.

Self-citations could be limited.

Comments on the Quality of English Language

The English language should be revised since there are some grammar errors and odd expressions that should be amended.

Round 2

Reviewer 3 Report

Comments and Suggestions for Authors

Although the Authors accepted some of the suggestions provided by this reviewer, other relevant revisions were not performed.

In particular:

1-A control group of wt A33-treated animals should have been performed for methodological soundness. I can understand contingent impediments, but this still represents a weakness point.

 2- The in vivo experiments on microglia depletion are still in the main text. As stated in my previous revision, since the experiment could not be performed due to collateral effects of the treatment, it does not provide information. Despite their potential relevance, the effects of the of the treatment on heart function are out of context.

 3-There are relevant concerns on how the Authors approached the issue of microglial role in PDE4B inhibition that are not adequately addressed in the manuscript.

The first paragraph of the results section, entitled “Long-term inhibition of PDE4B improves cognition in 8-month-old APPswe/PS1dE9 mice”, starts with the sentence “Activation of microglia has been shown to influence synapse number and affect cognition in AD”. Then, cognitive performances and synaptic density in the different experimental groups were the only parameters evaluated.

There is no evidence focused on microglia suggesting that, in the experimental setting here proposed, an effective microglial response (changes in microglial activation, proliferation, or others parameters) that follows PDE4B inhibition may be observed, thus justifying the focus on microglia.

Then, on primary microglial cultures the Authors test phagocytic activity and filopodia emission with and without PDE4B inhibition. Since there is an increase in filopodia emission after PDE4B inhibition, they conclude that this indicates increased surveillance and a more homeostatic phenotype for microglia, “possibly leading to less targeting and degrading of synapses”.  This second set of experiments seems untied to the previous, without a logical step connecting the two. Above all, they do not conclusively demonstrate that inhibition of PDE4B leads to reduced synaptic pruning.

It  is not clear why the Authors added, only in the response, experiments on microglia. A group of experiments is based on stimulation of BV2 cells- that cannot be compared to primary microglia- with LPS and another exploits Pde4b-ko microglia. They are not homogeneous with those presented in the manuscript and only partially face the criticism raised by this reviewer. In any case, they were not included in the main text.

Although the efforts are appreciable, in actual facts, the data presented in the manuscript still remain superficial and the conclusions, based on a reduced  synaptic pruning that follows PDE4B inhibition (not demonstrated), still overstate the results.

Round 3

Reviewer 3 Report

Comments and Suggestions for Authors

The revisions made by the Authors do not sufficiently face the criticism raised by this reviewer, as previously stated. Surprisingly, there is still a mention to the experiment that did not provide findings (lines 535-542).

Comments on the Quality of English Language

NA